# Enjoy Carefully: The Multifaceted Role of Vitamin E in Neuro-Nutrition

**DOI:** 10.3390/ijms221810087

**Published:** 2021-09-18

**Authors:** Liesa Regner-Nelke, Christopher Nelke, Christina B. Schroeter, Rainer Dziewas, Tobias Warnecke, Tobias Ruck, Sven G. Meuth

**Affiliations:** 1Department of Neurology, Medical Faculty, Heinrich Heine University Düsseldorf, 40225 Düsseldorf, Germany; christopherjannik.nelke@med.uni-duesseldorf.de (C.N.); christinabarbara.schroeter@med.uni-duesseldorf.de (C.B.S.); tobias.ruck@med.uni-duesseldorf.de (T.R.); meuth@uni-duesseldorf.de (S.G.M.); 2Department of Neurology, Klinikum Osnabrück, Am Finkenhügel 1, 49076 Osnabrück, Germany; rainer.dziewas@klinikum-os.de; 3Department of Neurology with Institute of Translational Neurology, University Hospital Münster, Albert-Schweitzer-Campus 1, 48149 Münster, Germany; tobias.warnecke@ukmuenster.de

**Keywords:** vitamin E, neurodegenerative diseases, nutrition, vitamin E supplementation, Alzheimer’s disease, personalized medicine

## Abstract

Vitamin E is often associated with health benefits, such as antioxidant, anti-inflammatory and cholesterol-lowering effects. These properties make its supplementation a suitable therapeutic approach in neurodegenerative disorders, for example, Alzheimer’s or Parkinson’s disease. However, trials evaluating the effects of vitamin E supplementation are inconsistent. In randomized controlled trials, the observed associations often cannot be substantiated. This could be due to the wide variety of study designs regarding the dosage and duration of vitamin E supplementation. Furthermore, genetic variants can influence vitamin E uptake and/or metabolism, thereby distorting its overall effect. Recent studies also show adverse effects of vitamin E supplementation regarding Alzheimer’s disease due to the increased synthesis of amyloid β. These diverse effects may underline the inhomogeneous outcomes associated with its supplementation and argue for a more thoughtful usage of vitamin E. Specifically, the genetic and nutritional profile should be taken into consideration to identify suitable candidates who will benefit from supplementation. In this review, we will provide an overview of the current knowledge of vitamin E supplementation in neurodegenerative disease and give an outlook on individualized, sustainable neuro-nutrition, with a focus on vitamin E supplementation.

## 1. Introduction

Vitamin E was first discovered by the American endocrinologist and anatomist Herbert. M. Evans, together with his assistant Katherine S. Bishop [1]. The isolated substance, later termed vitamin E, describes a group of compounds consisting of four tocopherol (TP)- and four tocotrienol (TT)-derivatives. They all share a chromanol ring as their structural basis and are therefore termed tocochromanols. The chromanol ring is hydroxylated in position 6 and, due to the methylation of the ring, α-, β-, γ- and δ-forms can be differentiated (Figure 1). TPs and TTs are classified according to their side chain: TPs contain one saturated fatty acid, whereas TTs contain a triple unsaturated fatty acid (Figure 1). The biological properties of vitamin E compounds are mainly determined by their structure. In humans, α-TP is the biologically most active form that binds with highest affinity to the α-tocopherol-transfer protein (α-TTP) [2]. α-TTP is a soluble protein found in the cytosol of hepatocytes in humans that acts to transport TPs between membrane vesicles, allowing for the distribution of vitamin E [3].

The definition of vitamin E and which derivates should be associated with this term are still under discussion. While vitamin E is extensively used as an umbrella term for different TP- and TT-forms, some authors use it as a synonym for α-TP. Azzi et al. justify the use of vitamin E as a synonym for α-TP by considering the classification of vitamins, which describes them as a group of substances essential for normal metabolism with deficiencies leading to disorders treatable by supplementation [4]. Accordingly, only α-TP should be named vitamin E, since it is the only form that has been shown to prevent the rare, inherited neurodegenerative disorder ataxia with vitamin E deficiency (AVED), which is caused by mutations in gene encoding for α-TTP, alpha tocopherol transfer protein (*TTPA*) [4,5]. However, since TPs and TTs are generally both implied when using the term vitamin E the scientific literature, we will also use vitamin E as an umbrella term in this review.

Considering the consequences of vitamin E deficiency—ataxia, dysarthria and neuromuscular disorders [6]—it is clear that this substance plays an important role in the central and peripheral nervous systems. Of interest, abetalipoproteinemia, a disorder characterized an inability to absorb fat and thereby profound the deficiency of chylomicrons, low-density lipoprotein (LDL) and very low-density lipoprotein (VLDL), all of which, being necessary for vitamin E absorption, result in ataxic neuropathy, retinal pigmentation, areflexia, and loss of proprioception [7]. Furthermore, equine neuroaxonal dystrophy/equine degenerative myeloencephalopathy, a neurodegenerative disorder affecting foals that resembles AVED, is associated with vitamin E deficiency, but is not associated with mutations in *TTPA*, such as is the case for AVED [8]. Interestingly, Kono et al. describe a case of juvenile spinocerebellar ataxia resulting from mutations in the phospholipid transfer protein (*PLTP*) gene, as well as *TTPA* [9].

Particularly in neurodegenerative disorders, vitamin E demonstrates notable benefits due to its antioxidant, anti-inflammatory and cholesterol-lowering properties [10]. Vitamin E supplementation as a therapy, particularly for neurodegenerative disorders, appears feasible and has been widely investigated, both in vitro and in vivo [11,12]. However, it has so far not been established in the prevention or treatment of these disorders, given the incoherent and sometimes contradictive results of interventional studies, and the apparent adverse effects of vitamin E supplementation [13]. Following recent insights into genetic polymorphisms, which play a crucial role in the metabolism of vitamin E, a new approach applying personalized medicine has emerged [14]. In this review, we will discuss the advantageous and disadvantageous effects of vitamin E in the context of neurodegenerative disease, as well as the factors that should be taken into consideration when tailoring personalized vitamin E supplementation strategies.

## 2. Characteristics of Vitamin E in the Context of Neurodegenerative Diseases

Structural differences give rise to biological variability among vitamin E derivatives (Table 1). These differences are important when considering the disease-modifying and preventive effects of supplementation.

α-TP has previously been in scientific focus due to its high bioavailability [49]. TPs are known to have antioxidant effects by increasing the activity of antioxidant enzymes and free radical scavenging. They can therefore interrupt free radical chain reactions. The free hydroxyl group on the aromatic ring is responsible for these antioxidant properties, as it is capable of scavenging free radicals, resulting in a relatively stable vitamin E radical. This radical can be reduced by ascorbic acid, which is then regenerated by glutathione [50].

Given its antioxidant properties, vitamin E has been considered an attractive therapeutic agent for the prevention and treatment of neurodegenerative diseases, such as Alzheimer’s (AD) and Parkinson’s disease (PD), where oxidative stress is an important pathophysiological driver [51,52]. Due to structural differences, TTs exert comparable or even more pronounced antioxidative effects, since they are distributed more homogeneously in the lipid membrane, and because recycling from chromanoxyl radicals is more efficient, thereby providing better reaction conditions, due to the stronger disordering of membrane lipids [53].

In addition, certain TPs and TTs exert anti-inflammatory effects through various methods of interference with the cellular and humoral immune systems. The antioxidative effects of TPs and TTs are intrinsically linked to their anti-inflammatory properties, since oxidative stress is a part of the inflammatory response [54]. However, certain effects are independent of these antioxidative properties. As such, vitamin E interferes with the inflammatory response at different levels, e.g., via transcription factors, signaling cascades, and the synthesis of signaling molecules [55]. α-, γ- and δ-TP have been shown to control Nrf-2 and NfkB signaling pathways in Caco-2 intestinal cells, which are crucial for the inflammatory response [56]. Of note, Jian et al. demonstrated that the suppression of prostaglandin synthesis through α-, γ- and δ-TP, by direct competitive inhibition of cyclooxygenase-2 (COX-2) with α-TP, inhibits tumor necrosis factor α (TNF-α) [57]. A meta-analysis of randomized controlled trials revealed a significant reduction of C-reactive protein (CRP) levels in groups supplemented with α- and γ-TP [58]. An additional meta-analysis suggested an association between α-TP-supplementation and decreased interleukin 6 (IL-6) serum levels [59].

However, the biological profile of vitamin E is not only restricted to the mediation of anti-inflammatory effects, but also characterized by immune-stimulatory properties, such as amplification of T-cell function [60]. Consistent with this, dietary in vivo studies suggested improved T-cell mediated functions, such as interleukin 2 (Il-2) production and T helper activity, in response to vitamin E supplementation [61,62].

Aside from these immunomodulatory effects, antineoplastic properties have been observed for vitamin E in case-control studies, as well as in interventional studies, but without reaching a definite conclusion [63,64,65]. Since most of these studies were performed with α-TP, the conflicting results shifted the focus to other vitamin E compounds. As such, a γ-TP-rich TP mixture demonstrated inhibitory potential on lung, colon, prostate, and breast cancer cell lines [22,23,25,66]. In mice with an induced lung tumor, application of the TP mixture resulted in reduced tumor burden, volume and multiplicity. Lower levels of DNA damage and thereby DNA repair, as well as increased apoptosis, were also observed [67]. Furthermore, a recent study suggested potent cytotoxic effects on brain cancer cells through γ-TT combined with the indole alkaloid jerantinine [24].

The cholesterol-lowering properties of the vitamin E compounds further demonstrate their use as therapeutic substances in chronic disease [68]. In 1986, Qureshi et al. isolated a cholesterol-lowering substance out of barley, later found to be α-TP [18]. In in vivo as well as human interventional studies, vitamin E compounds produced an improved serum cholesterol profile. This effect was partly attributed to the inhibition of HMG-CoA-reductase, which is unique to TPs. The comparison of different vitamin E compounds revealed a 30-fold higher inhibition of cholesterol biosynthesis for γ-TT compared to α-TP [27,69]. In human neuroblastoma cells, both α-TP and α-TT induced the reduction of total cholesterol as well as free cholesterol. Since α-TP achieved a more marked reduction of total cholesterol than free cholesterol, a link between this cholesterol-lowering effect and cholesterol esters is implied [28].

Aside from the modulation of de novo cholesterol synthesis at the protein level, vitamin E also affects gene transcription underlying cholesterol biosynthesis. Valastya et al. reported a reduction in the expression of genes responsible for cholesterol biosynthesis in hepatocytes upon α-TP treatment [19].

While all of the characteristics of vitamin E compounds discussed so far are indirectly neuroprotective due to the neurotoxic potential of inflammation [70] and oxidative stress [71,72], direct neuroprotective effects have also been described. In HT4 hippocampal cells, α-TT but not α-TP inhibited glutamate-induced pp 60 (c-Src) kinase activation and thereby cell death, with TTs being particularly potent [34]. Moreover, the modulation of 12-lipoxygenase, which also mediates glutamate-induced neurodegeneration, through α-TT was described, suggesting a further route for α-TT-mediated neuroprotection [35].

As such, epidemiological studies of patients aged 65 years or older reported that high dietary intake of vitamin E is inversely correlated with AD incidence. Interestingly, this effect was more pronounced with a combination of vitamin E compounds than with α-TP alone [73]. However, in patients with mild cognitive impairment (MCI) or AD, α-TP supplementation neither attenuated the progression of dementia nor improved cognitive function [12]. In comparison with donepezil, a standard therapy for symptomatic control of AD, the supplementation of 2000 IU vitamin E was of no benefit to patients with MCI, whereas donepezil was associated with a reduced progression of AD during the first 12 months of treatment [74]. A closer look into the bioavailability and metabolism of vitamin E is warranted for a potential explanation of this translational roadblock.

In PD, high dietary vitamin E, intake was inversely correlated with the occurrence of PD, independent of age or gender [11]. In a double-blind placebo-controlled trial with 60 PD patients, the dietary intake of 1000 mg omega-3-fatty acid and 400 IU vitamin E led to an improvement in the unified Parkinson’s disease rating stage (UPDRS), total antioxidant capacity and glutathione concentration compared to placebo. There was also a decrease in high-sensitivity C-reactive protein (hs-CRP) and favorable effects on markers of insulin metabolism [44]. However, a placebo-controlled clinical trial (deprenyl and tocopherol antioxidative therapy of parkinsonism (DARATOP)) treated patients with early PD with 10 mg/d depernyl and/or 2000 IU/d TP to investigate whether this supplementation strategy could extend the time until levodopa therapy is required. Depernyl displayed protective properties, whereas TP demonstrated no efficacy [45]. Interestingly, Fahn et al. reported an extension of 2.5 years to the time before requirement of levodopa therapy, due to the administration of α-TP and ascorbate in patients with early PD [75]. Contrasting these observations with those of epidemiological studies, which indicate that vitamin E may well be of benefit in PD, the inconsistent results of the interventional studies may explain why it has so far not been successfully integrated into clinical practice [76,77].

Few studies have investigated vitamin E for other neurodegenerative disorders such as Huntington’s disease (HD) or amyotrophic lateral sclerosis (ALS). There is evidence suggesting that the excessive activation of glutamate-gated ion channels followed by cell death through oxidative stress is the cause of HD pathogenesis [46]. Hence, the antioxidative properties of α-TP may be beneficial for HD patients. However, the high-dose treatment of 73 HD patients with α-TP demonstrated no overall effect on neurologic or neuropsychiatric symptoms, but a selective effect on neurological symptoms of patients with early HD [46].

Since vitamin E had favorable effects on the onset and progression of murine ALS, it was considered a treatment or additive therapy in the management of ALS [78]. In a double-blind, placebo-controlled study including 289 patients suffering from ALS for less than 5 years, the supplementation of α-TP had no effect on the deterioration of function, as assessed by the modified Norris limb scale. However, patients receiving α-TP were less likely to progress from state A to the more severe state B on the ALS Health State scale [48]. Another study investigating α-TP as an add-on therapy to riluzol in ALS reported no significant effect regarding survival rates, calculated time to death, tracheostomy, or permanent assisted ventilation [47].

## 3. Genetics in the Metabolism of Vitamin E

The metabolism and therefore bioavailability of vitamin E can be influenced by various factors, such as interaction with other nutritional compounds or pharmaceutics [49], gender [79], age [80], and lifestyle [81]. Due to the inhomogeneous outcome of interventional studies, individual response to vitamin E supplementation was considered as a potential explanation. Genetic heterogeneity arising from single nucleotide polymorphisms (SNPs) as a determinant of vitamin E homeostasis emerged as a hypothesis. SNPs are variations of a single nucleotide in a genome that can influence the biological properties of the encoded protein when occurring in coding regions [82,83]. Döring et al. described SNPs associated with genes that have a role in vitamin E metabolism, with the following genes considered as possible targets of SNPs, based on their function: lipoproteinlipase (*LPL*), tocopherol (α) transfer protein (*TTPA*), tocopherol-associated protein (*TAP*), multidrug resistance protein 2 (*MRP2*), pregnane X receptor (*PXR*), and the genes encoding cytochrome P450 enzymes (*CYP3A5, CYP3A4* and *CYP4F2*). In the exons of *TTPA, TAP* and *CYP3A5*, only a few coding SNPs (cSNP) were found. The cSNP frequency calculated was 503–837 bp per cSNP and is therefore not highly polymorphic. There is also a common SNP in *TAP*, which leads to an exchange of amino acids in the N-terminal functional domain of the protein. In *LPL, MRP2, PXR, CYP3A4*, and *CYP4F2*, cSNPs were reported with a range of 100 bp per cSNP, constituting a high number of polymorphisms [84].

Several genome-wide and candidate gene association studies have since uncovered further SNPs in proteins involved in vitamin E absorption efficiency or catabolism. This includes SNPs in *CYP4F2*, the gene encoding for cytochrome P4504F2, which catabolizes vitamin E, as well as scavenger receptor class B member 1 (*SCARB1*), which encodes scavenger receptor class B member 1, a plasma membrane receptor for high-density-lipoprotein (HDL), and the apolipoprotein *A1/C3/A4/A5*-gene cluster, which encodes for apolipoproteins that are associated with α-TP status [85,86,87]. Borel et al. showed that the postprandial chylomicronic α-TP response to TP-rich meals was highly variable among subjects. The interindividual variability in TP bioavailability was estimated at 81%, to which 28 SNPs and 11 genes were identified to be potentially contributing. Since most vitamin E is transported from the intestine to the liver and other organs by chylomicrons, 7 genes were involved in the postprandial chylomicronic triacylglycerol response. The other 4 genes were associated with the chylomicronic-TP response. The authors further observed that the plasma TP-concentration was positively correlated with the chylomicronic α-TP response to TP-rich meals, highlighting the importance of interindividual ability to respond to dietary tocopherol intake as an influential factor for α-TP serum concentration [88].

α-TTP plays an important role in vitamin E homeostasis [89]. Resultant from two SNPs in the *TTPA* gene are two variants, E141K and R59W, which are associated with ataxia due to vitamin E deficiency. They cause the reduced binding of α-TP to α-TTP in vitro, as the variants are located near the ligand-binding domain of *TTPA* [90]. These findings are in line with Wright et al., who showed an approximate 3% lower baseline vitamin E plasma level associated with the *TTPA* (−980T > A) variant in the promotor region of *TTPA* [91]. Regarding vitamin E catabolism, cytochrome P450 plays a substantial role, as it catalyzes the initial step in the vitamin E-ω-hydroxylase pathway, and notably contributes to vitamin E levels [92]. Bardowell et al. discovered two SNPs differently influencing enzyme activity. While the *CYP4F2* W12G variant leads to increased activity of the enzyme towards TPs and TTs, the *CYP4F2* V433M variant reduces enzyme activity for TPs, but not for TTs [93]. In a clinical trial comparing these genetic variants, the V433M genotype was associated with significantly higher plasma α-TP levels after 48 weeks of vitamin E supplementation (pioglitazone versus vitamin E versus placebo for the treatment of non-diabetic patients with nonalcoholic steatohepatitis (PIVENS) study: r = 20.35, *p* = 0.004 and treatment of nonalcoholic fatty liver disease in children (TONIC) study: r = 20.34, *p* = 0.026) [94].

## 4. Vitamin E Can Fuel the Pathogenesis of Neurodegenerative Diseases

Although vitamin E displays beneficial effects, not only in neurodegenerative disease, but also cancer, cardiovascular disease and infections, current evidence does not support vitamin E supplementation in the treatment or prevention of these diseases. This is due to inconsistent study results and the evidence that high-dose vitamin E supplementation (>400 U/d) might increase all-cause mortality [13]. Since these studies often involve malnourished populations or combined vitamin E with other substances, it remains unclear whether or not this effect would be seen in a population that is not nutritionally deficient.

Mechanisms explaining increased all-cause mortality are still debated. However, a number of adverse effects have been observed for vitamin E. Very high dosages of vitamin E (44 mg/kg body weight) increased blood pressure in spontaneously hypertensive stroke prone (SHRSP) rats [95]. There was also a rise in: phosphorylated neurofilament H protein, a prognostic marker of neurological disorders [96] and acute ischemic stroke [97]; glial fibrillary acidic protein associated with AD [98]; and cathepsin D in the CNS [95]. A study investigating the neuroinflammatory response following ischemic stroke in α-TTP-deficient mice treated with an α-TP diet (1680 IU/d) observed an exacerbation of ischemic stroke injury, due to supraphysiological brain injury accompanied by an increase in markers of oxidative injury and neurodegeneration [99].

Regarding AD, Grimm et al. investigated the in vitro effect of α-, γ- and δ-TP on the synthesis and degradation of amyloid β (Aβ), a peptide aggregating in extracellular plaques and a key driver of AD pathophysiology. Surprisingly, there was an increase in Aβ synthesis after incubation of SH-Sy5Y APP cells with 10 mM vitamin E over 24 h [100]. In contrast, Azzori et al. reported a reduction in Aβ42 concentration in SH-SY5Y APP Swe cells, due to incubation with α- and γ-TP [101]. Since Grimm et al. used an antibody that detects the last 5 amino acids of Aβ, and thereby not only Aβ42 but total Aβ, it may be that there is a distinct influence of vitamin E compounds on different Aβ forms. Further experiments by Grimm et al. highlighted that the increased Aβ synthesis is also due to increased γ- and β-secretase activity, especially through γ- and δ-TP. α-TP increased the protein levels of presenilin 1, a component of gamma secretase. The expression of genes encoding compounds of β-secretase was increased in response to α-TP [100].

In contrast, an in vivo study using mice showed no effect of vitamin E on the expression of β-site of APP cleaving enzyme (*BACE-1*) or a disintegrin and metalloproteinase domain-containing protein 10 (*ADAM-10*) [102]. In another study, the same group was able to validate the findings of increased Aβ synthesis through vitamin E compound supplementation [28]. Here, α-TP and α-TT amplified Aβ synthesis after incubation on SH-SH5Y cells. These effects were mild. However, given the long preclinical phase of AD, even a small change in pathogenesis could result in earlier manifestation of clinical symptoms. To test the influence of α-TT on a non-AD in vitro model, Grimm et al. used SH-SY5Y WT cells. Surprisingly, there was an even more pronounced effect on Aβ synthesis [28]. A possible explanation could be the increased substrate presence in SH-SY5Y APP cells, resulting in enhanced Aβ degradation. Given that high Aβ level sustained over a long period of time may increase Aβ catabolism, unaltered Aβ synthesis as observed by Arrozi et al. could be explained by the treatment period of over six months [101]. A study investigating TP levels in the human cortex observed an association between higher α- and γ-TP levels and lower total and activated microglia density in cortical regions, suggesting a microglia-mediated beneficial effect on the slowly accumulating AD neuropathology [21]. In later stages of AD, continued microglial activation can exacerbate tau pathology and negatively affect neurons and synapses [103,104]. During these stages, ameliorated microglial activation could be favorable. However, microglia activation may be protective in the early stages of AD, as microglia clear soluble Aβ, build protective barriers around Aβ plaques and remove debris [105,106]. Furthermore, a specific microglia cell type, disease-associated microglia (DAM), which has the potential to limit neurodegeneration, has been described [107]. Consequently, TPs being associated with reduced microglial activation highlights the possibility of further, perhaps adverse, effects of vitamin E supplementation on AD, as the transition into DAM might be restricted. On the contrary, in vitamin E-deficient mice, RNA-sequencing of the spinal cord demonstrated the upregulation of genes associated with the innate immune response, indicating that microglial activation may result from tocopherol deficiency [108].

Overall, this underpins the need for the proper timing of vitamin E supplementation in the course of neurodegenerative disease to improve the outcomes of treatment and mitigate adverse effects (Figure 2).

## 5. Is Vitamin E Supplementation Suitable for Everyone?

Data from randomized controlled trials suggests that there are beneficial properties of vitamin E in neurodegenerative disease, as it is associated with a reduced risk of disease development and may slow disease progression [10]. However, due to inconsistent interventional trails that failed to validate the proposed favorable findings, vitamin E is not yet a part of the treatment or prevention of these disorders [109]. In addition, the evidence suggesting high-dose vitamin E may increase all-cause mortality warrants further caution [13,110]. Clearly, vitamin E is not suitable as a therapy that is to be used indiscriminately. However, in the current scientific landscape, the beneficial effects of vitamin E supplementation seem to outweigh the possible adverse effects, the latter to be considered for individual cases. As already discussed for AD, vitamin E has been shown to increase Aβ-synthesis in an in vitro AD model [28].

Due to its anti-inflammatory, antioxidative and cholesterol-lowering properties, vitamin E is considered a suitable therapeutic or preventive strategy. In line with the framework of personalized medicine, it could be argued that patients with a high inflammatory response, cholesterol levels or the occurrence of oxidative stress may benefit from vitamin E supplementation. In contrast, patients with low cholesterol, minimal inflammatory burden and little oxidative stress may be adversely affected by supplementation. Regarding the effects of α- and γ-TP on activated microglial cell density [21], supplementation can lead to either desirable or detrimental effects, dependent on disease progression and method of administration.

When analyzing a patient’s suitability for supplementation, not only metabolic status and disease progression need to be taken into consideration, but also comorbidities and comedication. Vitamin E inhibits vitamin K-dependent coagulation factors (II, VII, IX and X) in the presence of vitamin K deficiency, and thereby induces coagulopathy [111]. This indicates an increased bleeding risk in patients taking vitamin K-dependent anticoagulation. Considering that falls are common in the elderly [112] and patients with neurodegenerative diseases [113], hemorrhagic complications are of high importance.

Lastly, when evaluating vitamin E supplementation, the administered dose needs to be defined. There is no consistent recommendation for the daily intake of vitamin E, as the recommended dietary allowance (RDA) is defined by different methods across countries. Furthermore, various individual factors influence the metabolism and thereby bioavailability of vitamin E (Figure 3) [49]. Considering the genetic polymorphisms in genes involved in the metabolism of vitamin E, the recommended intake should be tailored to the patient’s genetic profile.

## 6. Individualized, Proactive, Sustainable: The Future of Neuro-Nutrition

For many chronic disorders, current healthcare outcomes are considered inadequate [114]. Neurodegenerative diseases affect millions of patients worldwide [115]. Personalized medicine is emerging as a focus point in healthcare, aiming not only for better treatment, but also prevention, in order to improve outcomes and reduce the prevalence of chronic disease [116]. In this regard, nutrition and its beneficial role in the prevention and delay of neurodegenerative disease onset has become tremendously important in recent years. There is compelling evidence that inflammation and oxidative stress lead to the onset of several chronic diseases, and that diet might help to postpone, prevent or modulate the progression of these disorders [117].

Foods rich in vitamin E—e.g., wheat germ oil [118], almonds, hazelnuts and walnuts —have demonstrated protective properties in AD [119]. The Mediterranean diet, which is characterized by a high intake of fruit, vegetables, monounsaturated fat, fish, wholegrains, legumes, and nuts, is associated with a reduction in the risk factors for AD and cognitive decline. However, evidence for other neurodegenerative disorders and for markers of neurodegeneration is lacking [120]. Further studies are needed to evaluate which dietary components carry the greatest responsibility for the demonstrated effect, since similar properties have been described for different nutrients.

Interventional studies in humans were unable to substantiate the promise of the therapeutic potential of vitamin E regarding neurodegenerative diseases, as observed in in vivo studies [10]. This failure may be due to interindividual differences in vitamin E metabolism. To overcome this roadblock, an in-depth understanding of the factors affecting vitamin E metabolism is necessary to improve treatment strategies and, ultimately, treatment outcomes (Figure 4) [121]. Personalization of vitamin E supplementation is likely to be expensive but, given the immense burden imposed by neurodegenerative disorders on healthcare systems locally and globally [115], harnessing the therapeutic potential of vitamin E appears to be a worthwhile pursuit for future studies.

## Figures and Tables

**Figure 1 ijms-22-10087-f001:**
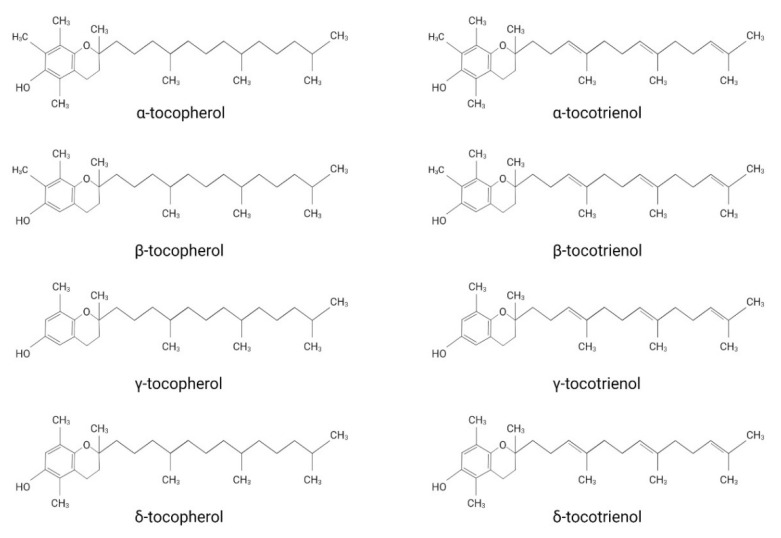
The structures of tocopherol- and tocotrienol- derivatives.

**Figure 2 ijms-22-10087-f002:**
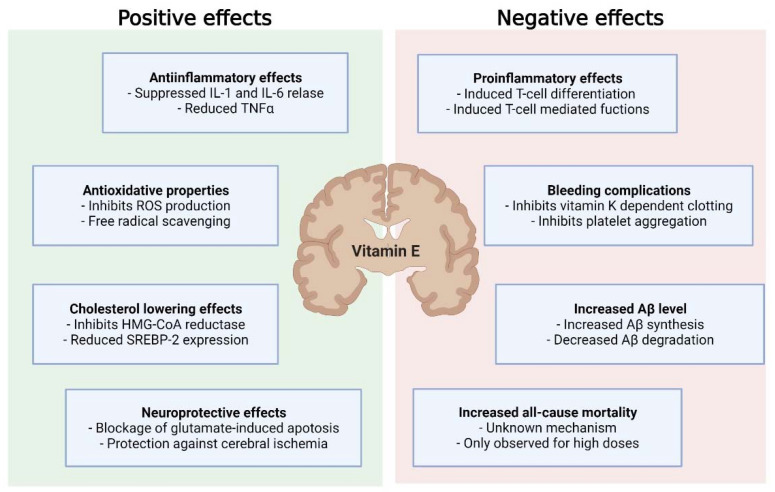
Positive and negative effects of vitamin E supplementation on neurodegenerative diseases. Abbreviations: IL-1, interleukin 1; IL-6, interleukin 6; TNF-α, tumor necrosis factor α; ROS, reactive oxygen species; HMG-CoA, β-Hydroxy-β-methylglutaryl-CoA; SREBP-2, sterol-regulytory-element-binding-protein-2; Aβ, Amyloid β.

**Figure 3 ijms-22-10087-f003:**
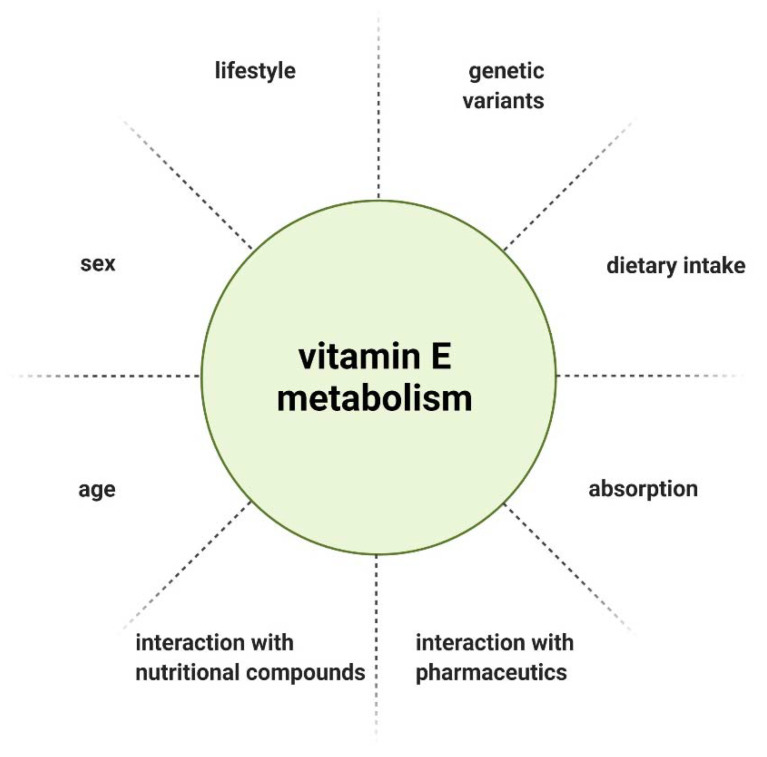
Factors influencing vitamin E metabolism.

**Figure 4 ijms-22-10087-f004:**
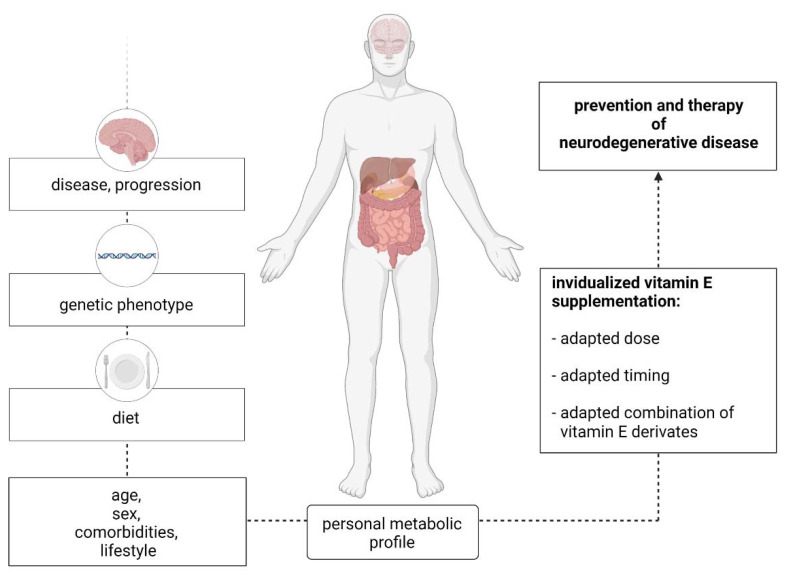
Individualized vitamin E supplementation and factors that should be considered beforehand.

**Table 1 ijms-22-10087-t001:** Characteristics of vitamin E-derivates.

Vitamin E		Biological Properties	Refrences
Tocopherols		Antioxidation	[15,16,17]
		Cholesterol-lowering	[18,19]
	α-Tocopherol	Anti-inflammation	[17,20,21]
	γ-Tocopherol	Anti-inflammation	[21,22]
		Anti-neoplastic	[23,24]
	δ-Tocopherol	Anti-neoplastic	[25]
Tocotrienols		Antioxidation	[15,26]
		Cholesterol-lowering	[27,28,29]
		Anti-inflammation	[16,30]
		Anti-neoplastic	[24,31,32,33]
	α-Tocotrienol	Immunostimulation	[34,35,36]
		Neuroprotection	[37,38]
	β-Tocotrineol	Neuroprotection	[39]
	γ-Tocotrienol	Neuroprotection	[38]
	δ-Tocotrienol	Immunostimulation	[40]

Given the prominent role of oxidative stress, chronic inflammation and dyslipidemia in the pathophysiology of neurodegenerative diseases, vitamin E is considered a promising therapeutic strategy (Table 2).

**Table 2 ijms-22-10087-t002:** Overview of relevant vitamin E-related clinical trials and results in neurodegenerative disorders.

Vitamin E Related Clinical Trials in Neurodegenerative Disorders					
	Subject	Supplementation	Duration	Results	Reference
Alzheimer’s disease	57 AD patients	800 I/dU vitamin E, Placebo	6 months	Differentiation responders/non-responders; responders showed lower oxidized glutathione levels than non-responders; cognitive status decreased in non-responders	[41]
	613 patients with mild to moderate AD	2000 IU/d α-tocopherol, 20 mg/d memantine, 2000 IU/d α-tocopherol + 20 mg/d memantine, Placebo	6 months	α-tocopherol compared with placebo resulted in slower functional decline	[42]
	341 moderate AD patients	10 mg/d monoamine oxidase inhibitor, 2000 IU/d α-tocopherol, Selegiline and α–tocopherol, Placebo	2 years	In patients with moderately severe impairment, α-tocopherol slows progression	[43]
Parkinson’s disease	60 PD patients	400 IU/d vitamin E + 1000 mg/d omega-3 fatty acids from flaxseed oil plus supplements, Placebo	12 weeks	Favorable effects on UPDRS, hs-CRP, TAC, GSH, and markers of insulin metabolism	[44]
	800 PD patients	2000 IU/d tocopherol +/or 10 mg/d deprenyl, Placebo	2 years	Tocopherol did not extend the time to levodopa therapy	[45]
Huntington’s disease	73 HD patients	3000 IU/d α-tocopherol	1 year	Selective therapeutic effect on neurologic symptoms for patients in the early course of the disorder	[46]
Amyotrophic lateral sclerosis	160 patients with either probable or definite ALS	5000 mg/d α-tocopherol, Placebo as add on to riluzol	18 months	No significant effect regarding survival rates, calculated time to death, tracheostomy, or permanent assisted ventilation	[47]
	289 ALS patients	500 mg α-tocopherol, Placebo as add on to riluzol	12 months	α-tocopherol group was less likely progressed from state A to more severe state B	[48]

Abbreviations: AD, Alzheimer’s disease; PD, Parkinson’s disease; HD, Huntington’s disease; ALS, Amyotrophic lateral sclerosis; UPDRS, unified Parkinson’s disease rating stage; hs-CRP, high-sensitivity C-reactive protein; TAC, total antioxidant capacity; GSH, glutathione.

## Data Availability

No new data were created or analyzed in this study. Data sharing is not applicable to this article.

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
