# Peer review of "Enjoy Carefully: The Multifaceted Role of Vitamin E in Neuro-Nutrition"

_ijms, 2021, doi:10.3390/ijms221810087_

Round 1
Reviewer 1 Report
Overall, this is an exceptionally well-written review that summarizes the current state of the literature regarding vitamin E supplementation in the prevention of neurodegenerative disorders. Both benefits and risks are thoroughly discussed. A few additional points are suggested below to strengthen the authors’ conclusions:
- To provide even further justification for the role of vitamin E in maintaining neural health, especially during postnatal development, the authors are encouraged to include comparative data that describe a similar phenotype to AVED across many mammalian species during deficiency (summarized in Muller DP 2010 Mol Nutr Food Res). A spontaneous familial disease resembling AVED, but not associated with polymorphisms in TTPA, exists in vitamin E deficient horses (Finno CJ, 2013;27(1):177-85). Additionally, the role for additional genetic polymorphisms contributing to AVED can also be further highlighted here (Kono, S. et al. J Neurol 2009;256(7):1180-1).
- Page 8: Discussion regarding microglia, vitamin E and AD. While the statement that microglia activation “may have protective effects in the early stages of AD…” is well-supported, the inferred conclusion from the de Leeuw F, et al. paper is not substantiated. With AD, homeostatic microglia transition to anti-inflammatory disease-associate microglia (DAM); a mechanism that is dysregulated with aging. With its anti-inflammatory role, vitamin E can maintain normal homeostatic microglia, poised to transition to DAM if needed. The results of the Leeuw F, et al paper suggest that tocopherols alleviated microglia activation, or maintained homeostatic microglia, in cortical brain regions. While this could be interpreted as not “allowing” for the transition to DAM, both arguments should be provided. In animal models, vitamin E deficiency upregulates DAM-associated transcripts in the CNS (Finno et al. 2018 Free Radic Mol Biol; 120:289-302). Thus, vitamin E sufficiency would likely maintain a homeostatic microglial state, with cells poised to shift to DAM if needed.
Minor Edits:
- Page 4: The Petersen, et al. RCT used 2000 IU (not 20,000) vitamin E
- Page 6: Another reference to summarize genes with polymorphisms that can affect vitamin E transport, etc: (Zingg, J. 2008 Nutr Rev 66(7):406-14).
Reviewer 2 Report
In the manuscript titled “Enjoy carefully: The multifaceted role of vitamin E in neuro[1]nutrition”, Authors aimed to review current knowledge on vitamin E supplementation in neurodegenerative diseases. This is an important topic because increasing the understanding the vitamin E supplementation could improve treatments and prevention of neurodegenerative diseases. The following comments need to be addressed in the presented study:
- Spelling and grammar need to be checked by native English-speaking scientist.
- The chapter 6. Individualized, proactive, sustainable: The future of neuro-nutrition needs to be expanded by discussing different vitamin E-rich diets/foods and their influence on neurodegenerative diseases. This will significantly increase the value of the review and increase an audience.
In conclusion, it is a valuable review.
